# Structure of Bovine CD46 Ectodomain

**DOI:** 10.3390/v15071424

**Published:** 2023-06-23

**Authors:** Hazel Aitkenhead, David I. Stuart, Kamel El Omari

**Affiliations:** 1Diamond Light Source (United Kingdom), Harwell Science and Innovation Campus, Didcot OX110DE, UK; hazel.aitkenhead@diamond.ac.uk; 2Division of Structural Biology, Wellcome Centre for Human Genetics, University of Oxford, Oxford OX37BN, UK; 3The Research Complex at Harwell, Rutherford Appleton Laboratory, Harwell, Didcot OX110FA, UK

**Keywords:** CD46, type I membrane protein, virus receptor, X-ray structure, pestivirus, BVDV, bovine viral diarrhoea virus

## Abstract

CD46, or membrane cofactor protein, is a type-one transmembrane protein from the complement regulatory protein family. Alongside its role in complement activation, CD46 is involved in many other processes, from T-cell activation to reproduction. It is also referred to as a pathogen magnet, because it is used as a receptor by multiple bacteria and viruses. Bovine CD46 (bovCD46) in particular is involved in bovine viral diarrhoea virus entry, an economically important disease in cattle industries. This study presents the X-ray crystallographic structure of the extracellular region of bovCD46, revealing a four-short-consensus-repeat (SCR) structure similar to that in human CD46. SCR1-3 are arranged linearly, while SCR 4 has a reduced interface angle, resulting in a hockey stick-like appearance. The structure also reveals the bovine viral diarrhoea virus interaction site in SCR1, which is likely to confer pestivirus specificity for their target host, CD46. Insights gained from the structural information on pestivirus receptors, such as CD46, could offer valuable guidance for future control strategies.

## 1. Introduction

CD46, also known as membrane cofactor protein (MCP), is a type I transmembrane complement regulatory protein (CRP) that was discovered in 1986 [1]. The complement system is part of the innate and adaptive immune responses that aim to protect the host upon pathogen invasion by swarming the pathogen with millions of complement activation fragments. It is a very powerful system and, therefore, requires tight regulation to prevent extensive host damage, which is the role of CRPs. The gene CD46 is found within the regulators of the complement activation gene cluster on chromosome 1q32, although it has been shown to be involved with many other processes alongside complement regulation [2].

The X-ray crystallographic structure of the human CD46 extracellular region [3] revealed that, like all other CRPs, it is composed of a number of short consensus repeats (SCRs) which are composed of around 60 amino acids arranged as a central beta-barrel surrounded by flexible loops. Between different SCRs, there is high sequence variability in these loops; however, four highly conserved cysteine residues form two disulphide bridges in each SCR, one at the top and the other at the bottom of the domain [3]. CD46 has four SCR domains, although the number found in other CRP family members varies from 4 to 30 [4]. In CD46, the SCRs are found in the extra-cellular portion at the N-terminus of the protein and make up the majority of the protein. Following the SCRs, there is a linker region abundant in serine, proline, and threonine residues (STP), a membrane-spanning anchor, and, finally, a cytoplasmic tail at the C-terminus of the protein. Splicing generates several distinct isoforms with changes in the STP region, resulting in varying O-glycosylation, as well as different cytoplasmic tails [5].

When responding to complement activation, CD46 binds the complement activation fragments C3b and C4b, which have been deposited on the host cell. CD46 then facilitates the cleavage and inactivation of these complement activation fragments by acting as a cofactor for serine protease factor 1 [1]. Alongside its role in complement regulation, CD46 has been linked to T cell activation, cell metabolism, reproduction, and autophagy, as well as being a pathogen receptor. Deficiencies in CD46 function have been shown to result in atypical haemolytic uremic syndrome, which is a rare thrombotic microangiopathy, and also in many other conditions, including several pregnancy-related disorders. In contrast, the upregulation of CD46 expression has been identified in all human cancers, although to differing degrees [6].

CD46 is termed a pathogen magnet due to the increasing number of pathogens that use it as a cellular receptor. Currently, these include 11 human pathogens (six types of bacteria and five different viruses), as well as members of the Pestivirus genus for bovine CD46 (bovCD46) and porcine CD46 (porCD46), and two bacterial pathogens in the case of teleost CD46 (teleostCD46) [7]. It is thought that CD46 is used as a cellular receptor by many pathogens due to its expression on almost all cell types. CD46’s complement regulatory activity, immune-modulating signalling functions, and internalisation mechanisms are also likely to play a large role in pathogenicity [6]. Different pathogens interact with different parts of the CD46 extracellular region, and the binding of pathogens to CD46 can result in changes to various cellular processes, potentially through changes to the CD46 cytoplasmic tail, such as cleavage [7]. These include changes in protein expression, cytoskeletal rearrangement, apoptosis, and immunosuppression [6].

Bovine CD46 acts as a cellular receptor for bovine viral diarrhoea virus (BVDV) [8], a bovine pestivirus causing reproductive losses, as well as other clinical signs, such as diarrhoea and mucosal disease in persistently infected animals [9,10]. This virus has high economic impact due to reduced reproductive performance, decreased milk production, depressed immune systems during recovery, and loss of animals [11]. As of 2019, the average direct costs of BVDV in Germany were estimated to be around 28.3 million Euros per million animals. Many countries, including Germany, have an active control program in place, but others, such as Spain, do not; indeed, it has been calculated that the complete eradication of the virus would not be economically beneficial [12]. Insights into the BVDV mechanism of cell entry are also likely to apply to other pestiviruses, such as atypical porcine pestivirus (APPV), as these have also been shown to utilise CD46 as a cellular receptor [13]. Pestiviruses are species-specific, and this has been shown to extend to CD46, with the expression of bovCD46 conferring increased susceptibility of porcine cells to BVDV [8]. It was shown that the deletion of the BVDV binding site on bovCD46 results in a vast reduction in BVDV cell entry events, but has little effect on cell surface motility [14], and that substituting six amino acids in the same domain dramatically reduced susceptibility to infection in a live calf [15].

High-resolution X-ray crystallography structures are available for the human CD46 SCR1-2 (PDB: 1CKL) [16] and SCR1-4 (PDB: 3O8E) [3], which have 41% and 48% sequence identity, respectively, with bovCD46 (Figure 1). Despite this sequence similarity, pestiviruses are sequence-specific and do not bind human CD46. Therefore, structural information on pestivirus receptors, such as CD46, could provide valuable insight into future control stratagems. This may become increasingly important as, alongside the well-known bovine and porcine versions, pestiviruses are now being found in an increasingly wide variety of animals, including bats and rats [17]. The primary objective of this study is to elucidate the experimental structure of bovCD46, which can be employed to gain insights into the mechanisms through which pestiviruses distinguish their host organisms.

## 2. Materials and Methods

### 2.1. Cloning and Expression

Protein expression was carried out as described elsewhere [20]. DNA encoding bovCD46 SCR1-2 (C43-I169) and SCR1-4 (C43-G295) was cloned into the transfer plasmid pHR-CMV-TetO2-Avi-His6, which adds an N-terminal secretion signal and a C-terminal Avi-His_6_ tag. This vector, alongside packaging and envelope plasmids (psPAX2 and pMD2.G), was transfected into HEK293T Lenti X cells to produce lentivirus. The lentivirus was used to transduce HEK293S GnTI-TetR cells to generate stable protein expression. The cells were cultured in DMEM media supplemented with 1% non-essential amino acids, 1% L-glutamine, and foetal bovine serum (10% during growth or 2% during expression/transduction). Expression was induced with the addition of doxycycline to 10 µg/mL. After seven days in roller bottles at 37 °C with 5% CO_2_ the media were harvested, centrifuged, filtered, and dialysed overnight against 20 mM Tris pH8 and 250 mM NaCl.

### 2.2. Purification and Crystallisation

Purification was carried out at room temperature. Dialysed media were passed over a nickel Sepharose IMAC gravity flow column (GE Healthcare, Little Chalfont, UK), washed with five column volumes of dialysis buffer with 20 mM imidazole, and eluted with five one-column volume elutions of dialysis buffer with 250 mM imidazole. Elution fractions were combined and concentrated using Amicon spin concentrations (Merck, Darmstadt, Germany) before final purification using size exclusion chromatography. A 16/600 Superdex s200 column (GE Healthcare, Little Chalfont, UK) was used with 10 mM Tris at pH 8 and 125 mM NaCl. Protein purity was analysed using SDS-Page gel electrophoresis before the pure fractions were combined and concentrated using Amicon spin concentrations (Merck, Darmstadt, Germany). The final protein concentration was calculated from the measured A280, molecular weight, and calculated extinction coefficient.

bovCD46_SCR1-2 was crystalised at 10 mg/mL in 0.1 M Tris at pH 8.5, 20% PEG 8000, and 0.2 M MgCl_2_ at 20 °C in sitting drop format (150 ηL drop with 1:1 protein: well solution ratio). bovCD46_SCR1-4 was mixed with Endo Hf (NEB, Hitchin, UK) at 2 units per µg of protein for 15 min on ice. The protein/Endo Hf mixture was then crystalised at 10.5 mg/mL in 0.1 M sodium cacodylate tetrahydrate at pH 6.4, 14% PEG smear broad, 0.2 M CaCl_2_, and 5% glycerol at 20 °C in the sitting drop format.

### 2.3. Data Collection, Structure Solution and Refinement

Crystal drops were flooded with a well solution containing either 25% ethylene glycol for bovCD46_SCR 1-2 or 20% glycerol for bovCD46_SCR 1-4 before flash cooling in liquid nitrogen. The datasets were collected at Diamond Light Source under beamline I03 and contained 3600 exposures with an omega rotation of 0.1°, and 0.025 s exposures were collected for bovCD46_SCR1-2; the beam had a flux of 9.2 × 10^11^ photons/s and 25% transmission was used. To collect 0.002 **s** exposures for bovCD46_SCR1-4 using a beam with a flux of 2.6 × 10^12^ photons/s, 100% transmission was used.

The datasets were processed with Dials [21] (bovCD46_SCR1-2) or autoPROC-STARANISO [22] (bovCD46_SCR1-4) via the diamond auto-processing pipeline [23]. Both bovCD46_SCR1-2 and bovCD46_SCR1-4 were solved using Phaser [24] with PDB: 3O8E used as a search model. Coot [25] and phenix.refine [26] were used to complete and refine the structures.

The data collection and refinement statistics can be found in Table 1. The structure factors and coordinates have been deposited in the PDB under accession code 8CI3 for bovCD46_SCR1-2 and 8CJV for bovCD46_SCR1-4.

## 3. Results

### 3.1. Structure Determination

The extracellular domains SCR1-2 (bovCD46_SCR1-2) and SCR1-4 (bovCD46_SCR1-4) of bovine CD46 were expressed in mammalian cells, as they are particularly suited for the expression of proteins that are heavily glycosylated and rich in disulphide bonds. The recombinant proteins were purified and crystallised under non-reducing conditions. bovCD46_SCR1-2 (glycosylation maintained) and bovCD46_SCR1-4 (deglycosylated in drop) constructs were successfully crystallised and diffracted to 2.3 Å and 2.8 Å resolutions, respectively (Table 1). The bovCD46_SCR1-2 and bovCD46_SCR1-4 structures were solved with molecular replacement using the equivalent human proteins as a search model. The bovCD46_SCR1-2 protein construct encompassed residues C43 to I169, and an additional threonine and glycine were resolved at the N-terminus resulting from the AgeI cloning site (Figure 2). The bovCD46_SCR1-4 construct covered residues C43 to G295, but the final glycine was not resolved in the structure, probably due to flexibility (Figure 2). One of the two monomers in the asymmetric unit only had electron density for SCR1-3—this was likely due to the interdomain flexibility in the structure. The structure revealed that the extracellular portion of bovCD46 adopted an extended conformation of 112Å in length. The STP region which is situated between the SCR and the transmembrane domain was not present in these protein constructs.

### 3.2. Overall bovCD46 Fold

As depicted in Figure 2, the bovCD46_SCR1-2 structure consisted of SCR1 and SCR2 arranged head to tail in a linear fashion, while bovCD46_SCR1-4 contained SCR1 through to SCR4. SCR1 and SCR2 were arranged as in the bovCD46_SCR1-2 structure and SCR3 continued this linear arrangement. There was a much-reduced angle between SCR3 and SCR4, leading to a bend in the structure, resulting in a hockey-stick-like appearance. As CD46 is a type I transmembrane helix, the shaft, composed of SCR1-3, will be distal to the membrane and the SCR4 blade will be located proximal to the membrane. The overall structure was likely to be fairly flexible, as the electron density was better defined at crystal contacts in the centre of the protein and deteriorated towards either end of the bovCD46_SCR1-4 construct.

The SCR region of the protein was located outside the cell in the extracellular environment and, therefore, contained N-linked glycosylation. BovCD46 was predicted to be glycosylated in several places: N92 in SCR1, N122 and N145 in SCR2, and possibly also N158 (SCR2) and N282 (SCR4). The electron density was well resolved for the N-linked glycans in the bovCD46_SCR1-2 structure and the glycosylation sites at N92 and N122 could be seen (Figure 2). The HEK293S cells were unable to generate complex glycosylation, resulting in high mannose oligosaccharides. Two N-acetyl glucosamine (NAG) residues could be seen at both sites, with some density suggested for a single mannose at position N122 in one monomer. Interestingly, the third glycosylation at N145 was not resolved in bovCD46_SCR1-2, but was seen in bovCD46_SCR1-4. The fourth and fifth predicted glycosylation sites in SCR2 and SCR4, respectively, were not visible in either structure. However, SCR4, which contained the fifth predicted glycosylation site, was the least well-resolved part of the structure due to its flexibility and the fact that it was not stabilised by crystal contacts. As this glycosylation site was conserved throughout different species (Figure 1), it was likely to be present. All the glycans were arranged along the concave face of the protein, leaving the convex side bare to the solvent.

### 3.3. Individual Domain Fold

The overall individual SCR domain structure for CRPs consisted of a central four-stranded beta barrel surrounded by a few small beta strands and linker loops. In standard SCR nomenclature, the central beta strands are named B, C, D, and E, and the smaller strands A, B’, D’, and E’ (Figure 1). In the structures presented in this study, not every SCR contained all eight beta strands, although some semblance of a central beta barrel was present in all. Upon the superposition of the four SCRs (Figure 3), the central beta structures were structurally well-conserved; however, there were differences in the loop conformation responsible for the high root-mean-square deviation (RMSD) values between Cα atom positions after superposition (RMSD of between 3.7 Å and 4.6 Å over the 60–64 residues in the domain).

On examination of the SCR loop conformations, there were two interesting differences found in SCR1 and SCR3. There was a loop shift in SCR1 in the D’D loop, which encompassed the putative BVDV binding site—this will be described later. An extension of the CD’ loop in SCR3 (Figure 3) affected the SCR3–SCR4 domain interface. This loop contained an insertion of five hydrophobic residues causing it to adopt an extended conformation protruding from the bottom of the domain. The protrusion perturbed the head-to-tail domain interface found between the other SCRs in the protein and caused the bend between SCR3 and SCR4.

Every SCR contained four cysteines which formed two disulphide bonds, as this region of the protein was located in the extra-cellular environment. The disulphide bonds were found at the top and the bottom of each SCR domain, as shown in Figure 2 and Figure 3. Cysteine 1 (C43 in SCR1) bonded to cysteine 3 (C89 in SCR1) and cysteine 2 (C72 in SCR1) bonded to cysteine 4 (C102 in SCR1) due to the antiparallel beta structure of the domains. In the bovCD46_SCR1-2 structure, we can see that, for one of these disulphide bonds, the electron density supported two different cysteine conformations, in one of which the disulphide bond was broken, likely due to radiation damage.

### 3.4. Putative BVDV Binding Site

The putative BVDV binding site was located in SCR1 and encompassed two peptides, E66-V69 and G82-L87 [8]. When mapped to the structure, these peptides were found on the C and D beta strands and D’D loop and formed a single binding site due to the antiparallel beta strand configuration (Figure 4). The first peptide had sequence EQIV69 in bovCD46 and other species carried a similar sequence, usually starting with a negatively charged amino acid, such as aspartic acid or glutamic acid. The second residue was generally a positively charged arginine, except for the bovine sequence, where it was a glutamine. The third residue was normally a hydrophobic valine or isoleucine, which differed only in a single methyl group. The final residue in this peptide was not well conserved. As the sequence was well conserved with similar residues, it was not surprising that the structure of this peptide was similar in both the human and bovCD46 structures (Figure 4). The second peptide had sequence GQVLAL87 in bovCD46 and showed some sequence conservation in the first three residues. The first and third residues were generally small non-polar residues, such as glycine, alanine, valine, or proline, whilst the second residue was a much bulkier leucine or glutamine. In contrast, the final three residues had no sequence conservation between human, bovine, porcine, and sheep CD46. The pattern of conservation was repeated at the structural level, when comparing the human and bovine structures. The first three residues had a similar conformation in both structures, but there was a loop shift covering the final three residues which comprised part of the D’D loop (Figure 4).

### 3.5. Crystallographic Dimer

An intriguing feature of these two crystal structures was the crystallographic dimer formation, which was identical in both protein constructs, despite them varying in size, unit cell, and space group (Figure 5). The PISA server [27] results indicated an area of interaction of 577 Å^2^ for bovCD46_SCR1-2 and 581 Å^2^ for bovCD46_SCR1-4. The interaction occurred at the top of SCR2 and the SCR1-2 interface for both copies of the protein. The higher-resolution bovCD46_SCR1-2 structure showed the possibility of nine hydrogen bonds being involved in this interface. Two of these were between SCR1 of chain A and SCR2 of chain B; the remaining seven bonds were between SCR2 of chain A and SCR2 of chain B. There were also five salt bridges present between the SCR2s of the different chains. In bovCD46_SCR1-4, the PISA server reported five hydrogen bonds and five salt bridges, all occurring between the SCR2 of both chains. Our attempts to verify that this crystallographic dimer was physiological were not conclusive.

## 4. Discussion

The structure of bovCD46_SCR1-2 presented here had a linear domain arrangement—this contrasted with the bent domain arrangement seen in the human CD46. Unbound structures of human SCR1-2 showed a bend between the two domains and some interdomain flexibility [16]. It was hypothesised that there was an extended conformation to the DE-loop of SCR2 which caused the bent conformation due to steric interference with residues from SCR1 [16]. Upon the solution of a two-domain ligand-bound structure which had a linear arrangement, it was postulated that the binding of the ligand caused the straightening of the protein [28]. As the bovCD46_SCR1-2 structure presented here had a linear arrangement, but had no ligand binding, it seemed more likely that this domain interface was inherently flexible, irrespective of the ligand-bound state.

In the overall structure, bovCD46_SCR1-4 resembled human CD46 SCR1-4, forming an elongated protein of over 100 Å with a hockey-stick-like shape. The deviation from the expected linear arrangement was caused by a bend between SCR3 and SCR4. The bend was due to a hydrophobic insertion in the CD’ loop of SCR3, first identified in the human 4-domain structure, which was also present in the bovine structure [3]. This hydrophobic insertion was not seen in other members of the CRP family, so its presence in both human and bovCD46 points to an important functional role. When examining the sequence of CD46 from multiple sequences, it can be seen that the CD’ loop insertion and, therefore, the bent conformation was present in all (Figure 1).

The bent conformation presented two main options for cell surface positioning, as proposed by Persson et al [3]. Either there was a bend in the STP region, causing SCR4 to lie almost parallel to the membrane and the rest of the protein to stick straight out, like an antenna, or SCR4 pointed straight up from the STP and membrane anchor, forcing the rest of the protein to lie almost parallel to the membrane [3]. It was suggested that the protein may adopt both conformations through separate isoforms with different STP regions.

BovCD46 has been identified as a receptor for BVDV due to increased permissibility on the overexpression of bovCD46 in previously non-infectable cells [29]. The area of interaction with BVDV was identified as two peptides on SCR1 [8], as seen in both bovCD46 structures; these form a single interaction interface. Comparing this region with the same region in the human structure revealed a shift in the DD’ loop. It has been shown that changing the sequence of interaction site residues to those of porCD46 causes a gain of function in porcine cells and a loss of function in bovine cells [8]. This difference in interaction site conformation is likely to confer the pestivirus specificity for their target host CD46.

Studies have also identified that all four bovCD46 SCRs are necessary for BVDV entry; however, the addition of up to six further SCRs had little effect on BVDV susceptibility [8]. Considering these facts, it could be construed that SCR1 must be a certain distance (i.e., four SCRs) from the plasma membrane to interact with BVDV. However, an alternative interpretation would be that each individual SCR plays an important biological role. Thus, SCR1 is involved in the BVDV interaction, whilst SCR3 and SCR4 and their interface cause the kink in the protein. Furthermore, if the SCR region lies on the plasma membrane, then additional C-terminally disposed SCRs would have little effect on BVDV binding, as they would not change the distance of SCR1 from the membrane.

To summarise, this study presents the X-ray structure of bovCD46, which serves as the host receptor for bovine viral Diarrhoea virus (BVDV). While sharing similarities with its human counterpart, this study has successfully pinpointed the precise location and arrangement of the residues involved in the interaction with BVDV on the bovCD46 structure.

## Figures and Tables

**Figure 1 viruses-15-01424-f001:**
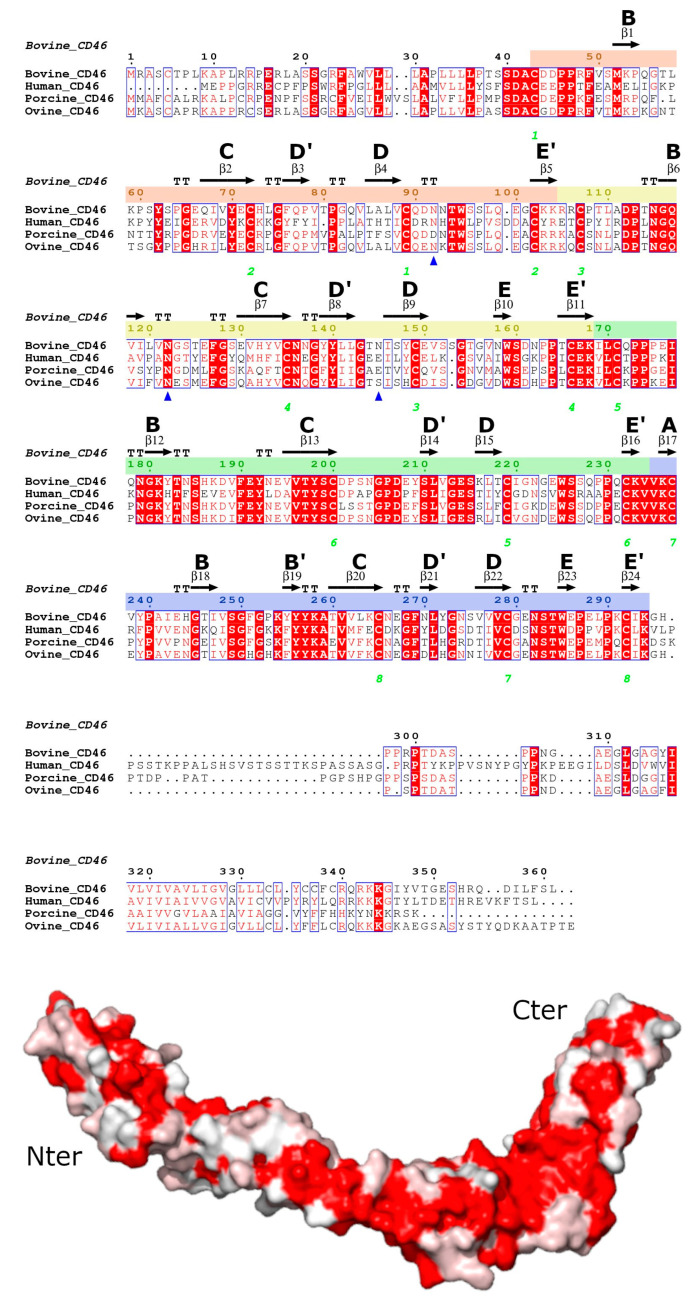
Amino acid conservation between CD46 proteins. Top—Sequence alignment of bovCD46 with CD46 from humans, pigs, and sheep generated using Clustal Omega [18] and ESPript web service [19]. Conserved residues are highlighted in red and similar residues are in red text. Secondary structure from bovCD46 is shown above and SCR beta sheet labelling has been added. The sequence numbering for each SCR has been highlighted as: SCR1—orange, SCR2—yellow, SCR3—green, and SCR4—blue. Green numbering below shows cysteines involved in disulphide bonds and glycosylated asparagines are marked with a blue triangle. GenBank accession numbers: Bovine— Q6VE48, Human—P15529, Porcine—O02839, and Ovine—W5PWS0. Bottom—Structure of human CD46_SCR1-4, with the surface representing sequence conservation on a scale of non-conserved white residues to conserved red residues.

**Figure 2 viruses-15-01424-f002:**
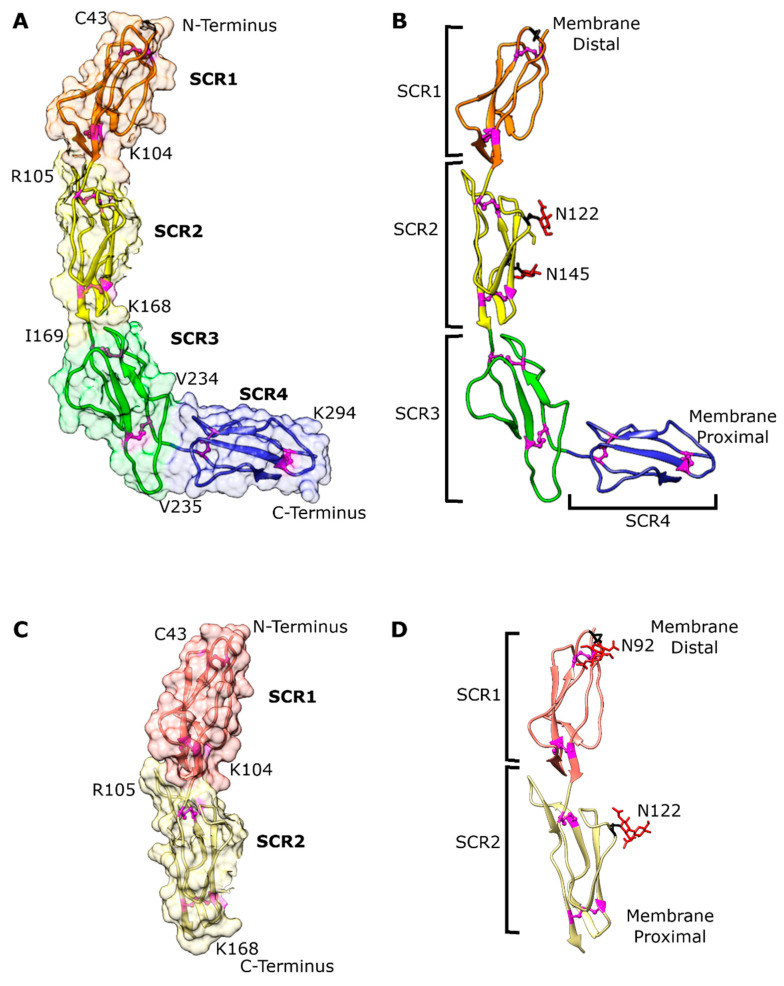
Overall structure of the bovCD46 extracellular domain. (**A**) Structure of SCR1-4 monomer in a drawing with the surface shown. SCR1 at the N-terminus is orange, SCR2 is yellow, SCR3 is green, and SCR4 at the C-terminus is blue. Cysteines and, therefore, disulphide bridges are shown in magenta balls and sticks. Domain boundaries are marked with the N-terminal boundary at the top left of the domain and the C-terminal at the bottom right. (**B**) Drawing of SCR1-4 monomer without surface with position in relation to marked membrane and labelled SCRs. The glycan residues are shown in red sticks and the asparagine residues, which are glycosylated in at least one monomer, are labelled and shown in black sticks. (**C**) As in (**A**), but for the SCR1-2 monomer. (**D**) As in (**B**), but for the SCR1-2 monomer.

**Figure 3 viruses-15-01424-f003:**
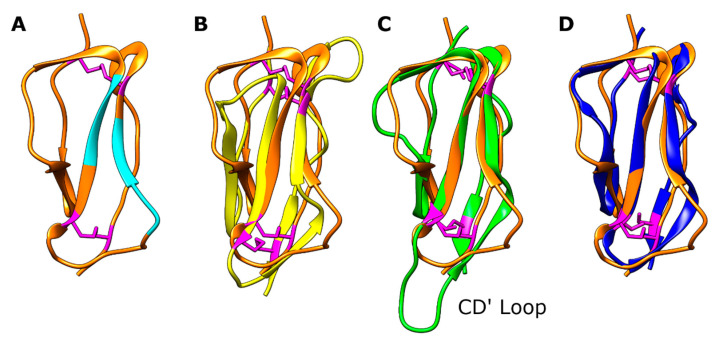
Structural comparison of individual SCR domains, SCR domain from bovCD46_SCR1-4 shown as a drawing with cysteines coloured magenta. (**A**) Drawing of SCR1 of bovCD46_SCR1-4 coloured in orange with the BVDV interaction site highlighted in cyan. (**B**) Drawing of SCR1 of bovCD46_SCR1-4 coloured in orange superposed with SCR2 in yellow, RMSD 4.1 Å over 60 residues. (**C**) Drawing of SCR1 of bovCD46_SCR1-4 coloured in orange superposed with SCR3 in green, RMSD 4.6 Å over 64 residues. (**D**) Drawing of SCR1 of bovCD46_SCR1-4 coloured in orange superposed with SCR4 in blue, RMSD 3.7 Å over 60 residues.

**Figure 4 viruses-15-01424-f004:**
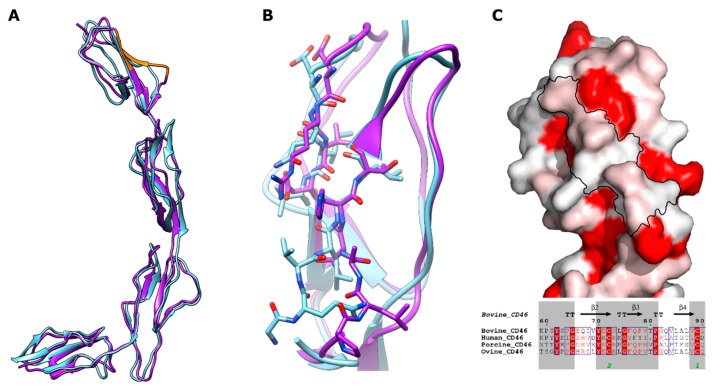
Structural comparison of bovCD46 and human CD46. (**A**) Drawing of bovCD46_SCR1-4 coloured purple with the BVDV interaction site highlighted in orange, a drawing of the human CD46 is superposed in blue (PDB:3O8E). (**B**) Closeup view of residues (shown as sticks) involved in the BVDV interaction site—the colour scheme is identical to (**A**). (**C**) Surface representation of bovCD46_SCR1-4 coloured by sequence conservation on a scale of non-conserved white residues to conserved red residues. BVDV interaction site is outlined in black and below is an excerpt from CD46 alignment focusing on BVDV interaction site peptides, both generated using ESPript web service [19].

**Figure 5 viruses-15-01424-f005:**
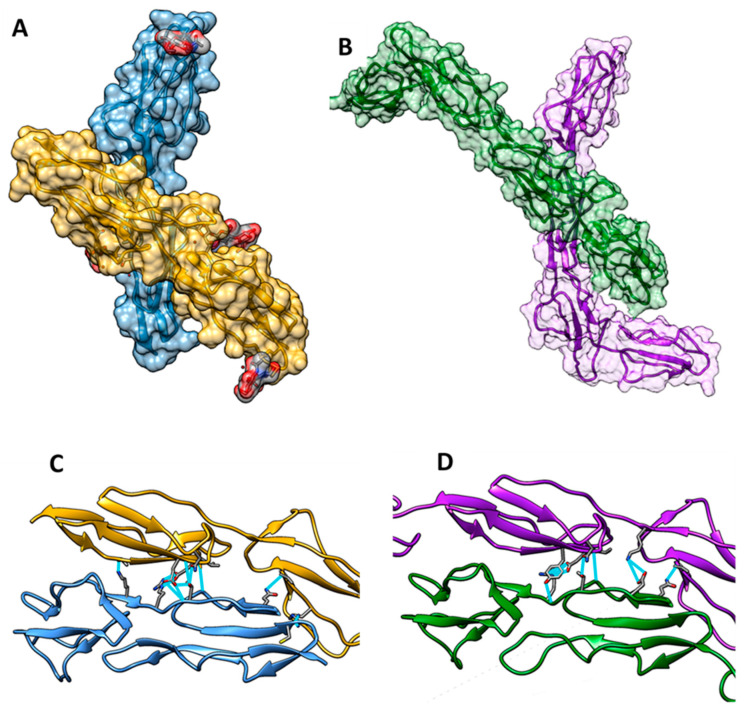
Crystallographic dimer found in bovCD46. (**A**) Model of bovCD46_SCR1-2 with both monomers from the asymmetric unit shown with a surface representation in blue and yellow. (**B**) Model of bovCD46_SCR1-4 with both monomers from the asymmetric unit shown with a surface representation in purple and green (the green monomer is a copy of the purple monomer superposed in the correct position, as SCR4 was not resolved in this chain). (**C**) Hydrogen bonding patch at the dimer interface of bovCD46_SCR1-2 with residues in hydrogen bonding distance shown in cyan sticks and the potential hydrogen bonds also in cyan. (**D**) Dimer interface of bovCD46_SCR1-4 with residues in hydrogen bonding distance shown in cyan sticks and the potential hydrogen bonds also in cyan.

**Table 1 viruses-15-01424-t001:** Data collection and refinement statistics for crystallographic structures (values for the outer shell are given in parentheses).

	bovCD46_SCR1-2	bovCD46_SCR1-4
Wavelength (Å)	0.98	0.98
Resolution range (Å)	44.12–2.33 (2.41–2.33)	91.73–2.84 (3.24–2.84)
Space group	*P* 4_1_ 2_1_ 2	*P* 4_1_ 2_1_ 2
*a*, *b*, *c* (Å)	91.09 91.09 121.12	129.73 129.73 120.19
α, β, γ (°)	90 90 90	90 90 90
Total no. of reflections	570,316 (55,868)	278,989 (11,165)
No. of unique reflections	22,417 (2168)	10,884 (544)
Multiplicity	25.40 (25.80)	25.60 (20.50)
Completeness (%)	99.40 (99.45)	Spherical: 44.1 (7.00) Ellipsoidal: 93.30 (73.30)
〈*I*/σ(*I*)〉	10.19 (1.12)	7.10 (1.80)
Overall *B* factor, Wilson plot (Å^2^)	52.03	50.37
*R* _merge_	0.23 (3.60)	0.49 (2.23)
*R* _meas_	0.24 (3.67)	0.50 (2.28)
*R* _pim_	0.05 (0.72)	0.10 (0.47)
CC1/2	1.00 (0.74)	1.00 (0.63)
Final *R*_work_	0.23 (0.35)	0.25 (0.39)
Final *R*_free_	0.26 (0.38)	0.29 (0.28)
RMSD bonds (Å)/angles (°)	0.01/0.90	0.02/1.84
Average *B* factors (Å^2^)	79.1	87.5
Ramachandran: favoured/allowed/outliers (%)	97.64/2.36/0.00	95.50/4.50/0.00
Rotamer outliers (%)	1.77	0.25
Clashscore	2.49	14.55
Number of TLS groups	4	7

## Data Availability

The structure factors and coordinates have been deposited in the PDB under the accession code 8CI3 for bovCD46_SCR1-2 and 8CJV for bovCD46_SCR1-4.

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
