# Peer review of "Structure of Bovine CD46 Ectodomain"

_viruses, 2023, doi:10.3390/v15071424_

Round 1

Reviewer 1 Report

The authors present a very interesting study that has determined the structure of the ectodomain of bovine CD46. The manuscript is well written and the conclusions drawn are well supported by the presented data.

I have made some comments below for the authors to consider.

The authors use bovine CD46 throughout the manuscript and the abbreviations bovCD46 or CD46bov are also used. I would suggest where "bovine CD46" is used for the first time the preferred abbreviation be introduced and then used for the remainder of the manuscript for consistency. 

I would suggest adding a sentence or two to the final paragraph of the introductory text that states the aim(s) of this study. Similarly, a corresponding statement at the end of the discussion stating the conclusion(s) of the study.

Line 15 - BVDV is an important contributor to disease across all cattle industries, suggest replacing "dairy industry" with "cattle industries"

Line 17 - suggest replacing "4" with "four" - similarly for other numbers less than 10 throughout the manuscript, as appropriate.

Line 17 The abbreviation "SCR" should be explained in full here for the abstract.

Line 19 The abbreviation "BVDV" should be explained in full here for the abstract. Though the term is only used once, an abbreviation is not required.

Line 22 Suggest using the full virus name as a keyword.

Line 27 suggest revision "of the innate and adaptive immune responses"

Line 69 suggest replacing "symptoms" with "clinical signs"

Line 92 Please provide appropriate references for bats and rats.

Line 117 suggest replacing "u" with the correct symbol for "micro"

Author Response

The authors present a very interesting study that has determined the structure of the ectodomain of bovine CD46. The manuscript is well written and the conclusions drawn are well supported by the presented data.

We would like to thank the reviewer for his/her positive comments and careful evaluation of the manuscript.

The authors use bovine CD46 throughout the manuscript and the abbreviations bovCD46 or CD46bov are also used. I would suggest where "bovine CD46" is used for the first time the preferred abbreviation be introduced and then used for the remainder of the manuscript for consistency. 

In the main text, bovine CD46 is cited for the 1st time, line 61, we have then changed all abbreviations to ‘bovCD46’ through the manuscript for consistency.

I would suggest adding a sentence or two to the final paragraph of the introductory text that states the aim(s) of this study. Similarly, a corresponding statement at the end of the discussion stating the conclusion(s) of the study.

At the end of the introduction, the following sentence has been added to clarify the aim of the paper (line 95): ‘The primary objective of this study is to elucidate the experimental structure of bovCD46, which can be employed to gain insights into the mechanisms through which pestiviruses distinguish their host organisms.’

At the end of discussion, the following sentence has been added: ‘To summarize, this study presents the X-ray structure of bovCD46, which serves as the host receptor for Bovine Viral Diarrhea Virus (BVDV). While sharing similarities with its human counterpart, the study has successfully pinpointed the precise location and arrangement of the residues involved in the interaction with BVDV on the bovCD46 structure.

Line 15 - BVDV is an important contributor to disease across all cattle industries, suggest replacing "dairy industry" with "cattle industries"

We agree with the reviewer comment and have changed "dairy industry" to "cattle industries" in the abstract (line 16).

Line 17 - suggest replacing "4" with "four" - similarly for other numbers less than 10 throughout the manuscript, as appropriate.

As requested by the reviewer, we have written all numbers less than 10 as letters throughout the text, but we have kept the numbers that identify a protein residue (for example ‘cysteine 1’).

Line 17 The abbreviation "SCR" should be explained in full here for the abstract.

We have added the definition of SCR in the abstract ‘short consensus repeat (SCR)’.

Line 19 The abbreviation "BVDV" should be explained in full here for the abstract. Though the term is only used once, an abbreviation is not required.

We have removed the abbreviation from the abstract as it is cited only once and replaced it with the full name: ‘Bovine Viral Diarrhea Virus’.

Line 22 Suggest using the full virus name as a keyword.

We have added ‘Bovine Viral Diarrhea Virus’ as a keyword (line 19).

Line 27 suggest revision "of the innate and adaptive immune responses"

This sentence has been corrected as requested (line 29).

 Line 69 suggest replacing "symptoms" with "clinical signs"

This sentence has been corrected as requested (line 72).

Line 92 Please provide appropriate references for bats and rats.

A reference to the paper by Wu Z. et al ([17]) has been added (line 95).

Line 117 suggest replacing "u" with the correct symbol for "micro"

"u" has been replaced with symbol for "µ" in the material and method section (line 122, 138).

Reviewer 2 Report

In this manuscript, authors tried to resolve the crystal structure of ectodomain of CD46, which can serve as pathogen receptors, including BVDV, T-cell activation etc.  They tried to compare, structure wise, with that of the human counterpart.

Just out of curiosity, if the overall structure presented in Figure 2 is correct, then there is no way that in reality, in Figure 3, SCR1 can superpose with SCR2, SCR3, or SCR4.  Please explain  why you want to do this analysis. Or delete it.

Enlarge Figure 1. It deserves.

Discuss the possible reasons why 2 isoforms of crystal dimers happened.  Could it be due to the lentivirus expression etc.

Minor points:

consistent wording on: SCR1-4 versus SRC1-SCR4, SCR1-2 versus SCR1-SCR2.

consistent wording on: rmsd versus r.m.s.d versus RMSD. Provide the full spelling of what is it.

Author Response

In this manuscript, authors tried to resolve the crystal structure of ectodomain of CD46, which can serve as pathogen receptors, including BVDV, T-cell activation etc.  They tried to compare, structure wise, with that of the human counterpart.

We would like to thank the reviewer for his/her comment and evaluation of the manuscript.

Just out of curiosity, if the overall structure presented in Figure 2 is correct, then there is no way that in reality, in Figure 3, SCR1 can superpose with SCR2, SCR3, or SCR4.  Please explain why you want to do this analysis. Or delete it.

We have superimposed SCR1 on SCR2,3,4 to highlight the structural differences between these SCR domains, for example the CD’ loop as shown in Figure 3C, responsible for the protein curvature. This is a common practice for structure comparison, but this does not imply that this is the ‘reality’.

Enlarge Figure 1. It deserves.

To address the reviewer comment, we have moved the structure at the bottom of the sequence alignment to enlarge the figure.

Discuss the possible reasons why 2 isoforms of crystal dimers happened.  Could it be due to the lentivirus expression etc.

As mentioned in text, we have solved the structure of 2 different constructs of bovine CD46: SCR1-2 and SCR1-4. These proteins were both expressed in HEK cells, but because their length is different, they crystallised in different crystallisation conditions as well as space groups and unit cells. Despite this, the same dimers interface are found in both crystals.

Minor points:

consistent wording on: SCR1-4 versus SRC1-SCR4, SCR1-2 versus SCR1-SCR2.

We thank the reviewer for spotting these inconstancies, we changed to a common nomenclature SCR*-* throughout the text.

consistent wording on: rmsd versus r.m.s.d versus RMSD. Provide the full spelling of what is it.

The inconstancy has been corrected and all ‘rmsd’ have been changed to ‘RMSD’ (Table 1, line 232, 233, 240, 242, 243). The full spelling is given line 232: ‘root mean square deviation’.

Reviewer 3 Report

The paper is overall very good. 

1. Minor revisions need to do, for example, many units ug/ml, nl.

2. Is it should be 3.7 and 4.6Å or 3.7Å and 2.8Å? And check the notes in Figure 5: (A), (B)-, (C), (D)-.

3. Uppercase letters and lowercase letters should be unified according to the guideline or the latest issue.

4. The journal names are not unified in the REFERENCES, then PLEASE check their abbreviations or full names.

English writing is very well.

Author Response

The paper is overall very good.

We would like to thank the reviewer for his/her comment and evaluation of the manuscript.

Minor revisions need to do, for example, many units ug/ml, nl.

We thank the reviewer for spotting these inaccuracies, we have changed the units: ‘u’ to ‘µ’   and ‘n’ to ‘η’ in the material and method section (line 122, 137, 138).

Is it should be “3.7 and 4.6Å” or “3.7Å and 2.8Å”? And check the notes in Figure 5: (A), (B)-, (C), (D)-.

We have added the Angstrom unit (line 233): “3.7 Å and 4.6Å” and in the Figure 5 legend, we have removed the hyphen after (B) and (D).

Uppercase letters and lowercase letters should be unified according to the guideline or the latest issue.

Section and sub-section titles have been corrected by using uppercase letters for each word.

The journal names are not unified in the REFERENCES, then PLEASE check their abbreviations or full names.

We thank the reviewer for spotting these mistakes, it turns out that Endnote was not using the right abbreviations, this is now corrected.